# Promising Prognosis Marker Candidates on the Status of Epithelial–Mesenchymal Transition and Glioma Stem Cells in Glioblastoma

**DOI:** 10.3390/cells8111312

**Published:** 2019-10-24

**Authors:** Yasuo Takashima, Atsushi Kawaguchi, Ryuya Yamanaka

**Affiliations:** 1Laboratory of Molecular Target Therapy for Cancer, Graduate School of Medical Science, Kyoto Prefectural University of Medicine, Kyoto 602-8566, Japan; ytakashi@koto.kpu-m.ac.jp; 2Center for Comprehensive Community Medicine, Faculty of Medicine, Saga University, Saga 849-8501, Japan; akawa@cc.saga-u.ac.jp

**Keywords:** glioblastoma, glioma stem cell, epithelial-mesenchymal transition, multivariable analysis, prognosis prediction

## Abstract

Multivariable analyses of global expression profiling are valid indicators of the prognosis of various diseases including brain cancers. To identify the candidates for markers of prognosis of glioblastoma, we performed multivariable analyses on the status of epithelial (EPI)–mesenchymal (MES) transition (EMT), glioma (GLI) stem cells (GSCs), molecular target therapy (MTT), and potential glioma biomarkers (PGBs) using the expression data and clinical information from patients. Random forest survival and Cox proportional hazards regression analyses indicated significant variable values for *DSG3*, *CLDN1*, *CDH11*, *FN1*, *HDAC3/7*, *PTEN*, *L1CAM*, *OLIG2*, *TIMP4*, *IGFBP2*, and *GFAP*. The analyses also comprised prognosis prediction formulae that could distinguish between the survival curves of the glioblastoma patients. In addition to the genes mentioned above, *HDAC1*, *FLT1*, *EGFR*, *MGMT*, *PGF*, *STAT3*, *SIRT1*, and *GADD45A* constituted complex genetic interaction networks. The calculated status scores obtained by principal component analysis indicated that GLI genes covered the status of EPI, GSC, and MTT-related genes. Moreover, survival tree analyses indicated that MES^high^, MES^high^GLI^low^, GSC^high^GLI^low^, MES^high^MTT^low^, and PGB^high^ showed poor prognoses and MES^middle^, GSC^low^, and PGB^low^ showed good prognoses, suggesting that enhanced EMT and GSC are associated with poor survival and that lower expression of EPI markers and the pre-stages of EMT are relatively less malignant in glioblastoma. These results demonstrate that the assessment of EMT and GSC enables the prediction of the prognosis of glioblastoma that would help develop novel therapeutics and de novo marker candidates for the prognoses of glioblastoma.

## 1. Introduction

The World Health Organization (WHO) classifies gliomas into four categories based on malignancy and overall survival (OS) [1]. Glioblastoma is the most malignant form of astrocytoma that is fast-growing (grade 4 glioma) [1,2]. The median OS of glioblastoma is 9–15 months and the five-year survival rate remains less than 5% [1,3,4,5]. Radiotherapy with six cycles of concomitant temozolomide, an oral alkylating agent with minimal additional toxicity, is the standard treatment after surgery in glioblastoma patients [2,4,5]. Thus, there is an immediate requirement for the early diagnosis and precise prediction of the prognosis for treatments of glioblastoma.

The invasiveness of glioblastoma depends on its high infiltration potential to invade the basement membranes of surrounding tissues [6,7,8,9]. Glioma cells are reprogrammed to have increased motility via weakened cell adhesions and a dysregulated cytoskeleton; this is known as epithelial–mesenchymal transition (EMT) [10]. EMT is a biological process that polarized epithelial cell sheets undergo, wherein multiple biochemical changes culminate in mesenchymal phenotypes [6]. Cells display altered morphology, resistance to chemotherapy, and *anoikis* (a form of programmed cell death of cells detached from the basement membranes) [11,12]. EMT is also involved in other biological processes, including implantation and embryo development, wound healing, tissue regeneration, and neoplasia associated with cancer progression [12]. During invasion in cancer, phenotypes also change as EMT is mainly induced by hypoxia and the transforming growth factor (TGF)-β released from glioma stem cells (GSCs), mesenchymal stem cells (MSCs), and myeloid cells recruited by hypoxia [13]. The reverse is also essential for the formation of distant or disseminated tumor nodules [10].

Stem cells, including cancer stem cells (CSCs), are pluripotent and capable of self-renewal, like induced pluripotent stem (iPS) cells induced by the transcription factors Oct3/4, Sox2, c-Myc, and Klf4 [14] or OCT4, SOX2, NANOG, and LIN28 [15]. The expression of OCT4, MYC, and KLF4 increases with increasing malignancy in astrocytomas, whereas MYC expression slightly decreases in recurrent glioblastomas [16]. Exposure to temozolomide (TMZ) increases the expression of KLF4 and reduces the expression of Nanog and OCT4 in glioma cells [16], indicating that stem cell factors, especially KLF4, play pivotal roles in GSCs. The most common GSC marker CD133, also known as prominin-1 (*PROM1*), is used to isolate GSCs by fluorescence-activated cell sorting (FACS) in primary glioma tissues and the cell lines U87 and T98G [17]. There are changes in metabolism in glioma and oxidative phosphorylation, glycolysis, and glutaminolysis replace redox balance, bioenergetics, and biosynthesis, respectively [18]. Tumor progression is enhanced by inflammation, which is an emerging hallmark of cancers that depends on the balance of glioma cell proliferation, migration, and escape from the immune system [18]. However, the correlation between EMT and GSCs in glioblastoma has not been reported to date.

In this study, we focused on EMT and GSCs and performed multivariable analyses using the expression data and OS in patients with glioblastoma. To determine the distribution of survival of glioblastoma patients, we composed prognosis prediction formulae for the following components: epithelium (EPI), mesenchyme (MES), glioma (GLI), GSC, molecular target therapy (MTT) genes, and potential glioma biomarkers (PGBs). Consequently, several candidate genes were identified as promising glioblastoma predictors.

## 2. Materials and Methods

### 2.1. Data Set

Expression data and clinical information of patients with glioblastoma deposited in The Cancer Genome Atlas (TCGA) and the Chinese Glioma Genome Altas (CGGA) were used [19,20]. Gene expression values of fragments per kilobase of exon per million reads mapped (FPKM) were normalized and subjected to the following analyses. The analyses were performed on the TCGA data set (151 samples) and the results were validated using the CGGA data set (135 samples) (Appendix A). Tumor specimens were collected from untreated patients during surgery.

### 2.2. Gene Annotation

Genes of interest were annotated online at GOstat2.5 (http://gostat.wehi.edu.au/) [21] and Database for Annotation, Visualization, and Integrated Discovery (DAVID)6.8 (https://david.ncifcrf.gov/) [22].

### 2.3. Clustering Analysis

Gene expression values were modified with the valiances from the means of patients, followed by hierarchical clustering analysis with the Ward method using the JMP built-in module (SAS Institute Inc., Tokyo, Japan). Heat maps were drawn to visualize their expression with a color configuration with red as high and green as low [23].

### 2.4. Random Survival Forests Analysis

Random survival forests analysis was performed to determine the variable importance factors distinguishing gene expression associated with patient survival using the randomForestSRC package in R [24,25,26]. The variable importance values reflected the relative contribution of each variable to the prediction for the survival time, which was estimated by randomly permuting the values and recalculating the predictive accuracy of the model.

### 2.5. Cox Hazards Regression Analysis

Correlations between gene expression and survival times were evaluated by the Cox proportional hazards regression analysis using the JMP built-in module (SAS Institute Inc.) [25,27].

### 2.6. Survival Analysis

The Kaplan–Meier analysis was performed to estimate the survival distributions of subgroups with the log-rank test using the JMP built-in module (SAS Institute Inc.). The hazard ratio (HR) and confidence interval (CI) were calculated with a logistic regression model according to patients’ survival times, which were assessed with a stepwise selection to compare the subgroups. Overall survival (OS) was defined as the date of diagnosis of glioblastoma to the date of death or last follow-up [19,20].

### 2.7. Graphical Lasso Estimation

Genetic interactions with hub networks among variables from gene expression were analyzed by the graphical lasso estimation of Gaussian graphical models, such as a sparse inverse covariance matrix using a lasso (L1) penalty, using the glasso package in R [28,29].

### 2.8. Principal Component Analysis (PCA)

Score correlation analysis was performed with formulae constituted by the first principal components derived from PCA. PCA was performed with the normalized FPKM values using the JMP built-in module (SAS Institute Inc.), which was used to classify the patients into the subgroups to estimate their prognoses in simple forms as linear combinations of expression values of the genes of interest [26].

### 2.9. Survival Tree Analysis

Tree-structured survivals were analyzed to identify the largest differences in the Kaplan–Meier survival curves by dividing the patients into the most appropriate subgroups, harboring variable spaces as overall survival and interval censoring according to a constant model of the response variable in each partition [30]. The survival tree analysis was performed using the rpart package in R [31].

### 2.10. Statistics

Statistical analyses were performed using R [32], Bioconductor [33], JMP10 (SAS Institute Inc.), and Microsoft Excel (Microsoft Corporation). *p* < 0.05 was considered statistically significant.

## 3. Results

### 3.1. Status of EMT and GSC Gene Expression in Glioblastoma

The aim of this study is to identify promising prognosis marker candidates of glioblastoma. As glioma is caused by EMT and maintained by glioma stem cells, in this study, we used the expression data and clinical information from 286 glioblastoma patients from TCGA and CGGA (Appendix A) and focused on representative genes involved in EMT and glioma (Figure 1, Appendix A) [34,35,36,37]. Expression of the 126 genes in 151 samples from the TCGA data set was shown with hierarchical clusters represented by heatmaps, including those in EPI (14 genes; Figure 1a), MES (23 genes; Figure 1b), GLI (53 genes; Figure 1c), GSCs (17 genes; Figure 1d), MTT (34 genes; Figure 1e), and PGBs (24 genes; Figure 1f). However, distinct expression patterns capable of stratifying the patients into clusters were not identified, suggesting a possibility that differential expression of the genes requires combinations to evaluate the prognoses of glioblastoma patients. Thus, we tested whether combinations of the genes were associated with EMT and GSC in glioblastoma.

### 3.2. Importance of Significant Variables and Hazard Ratios of the Genes

Random survival forest analysis indicated the variable importance of each gene and the Cox proportional hazards regression analysis showed significant coefficient and hazard ratios (HRs) (Appendix A). A high variable associated with a significant coefficient and HR was seen in the following: *DSG3* and *CLDN1* as EPI markers (Appendix A); *CDH11* and *FN1* as MES markers (Appendix A); *FLT1*, *HDAC3/7*, and *PTEN* as GLI markers (Appendix A); *L1CAM*, *OLIG2*, *BMI3*, and *STAT3* as GSC markers (Appendix A); *EGFR*, *HDAC1/3/7*, and *SIRT1* as MTT genes (Appendix A); and *TIMP4*, *IGFBP2*, *GFAP*, *GADD45A*, *MMP9*, and *MGMT* as PGBs (Appendix A). Increased expression of *DSG3* (HR 1.36, 95% confidence interval (CI) 1.11–1.67, *p* = 0.004), *FN1* (HR 1.29, 95% CI 1.08–1.56, *p* = 0.006), *IGFBP2* (HR 1.18, 95% CI 1.04–1.35, *p* = 0.011), *CLDN1* (HR 1.24, 95% CI 1.05–1.46, *p* = 0.013), *HDAC7* (HR 1.36, 95% CI 1.04–1.77, *p* = 0.023), and *L1CAM* (HR 1.10, 95% CI 1.01–1.19, *p* = 0.024) were associated with poor prognoses of glioblastoma (Appendix A). In contrast, an increased expression of *TIMP4* was associated with good prognosis of glioblastoma (HR 0.88, 95% CI 0.79–0.98, *p* = 0.025; Appendix A). An increase in *MGMT* and *MMP9* levels showed slightly poor prognoses of glioblastoma (*p* > 0.05) (Appendix A).

### 3.3. Evaluation of the Formulae for the Prediction of Prognosis of Glioblastoma

To determine patient survival, formulae were composed for 22 genes (Appendix A) in a simple linear form using the sum of the integration of the coefficients and fragments per kilobase of exon per million mapped fragments (FPKM) of the normalized genes (Appendix B). Glioblastoma patients were divided into two subgroups based on the median status scores using the formulae. Subsequently, Kaplan–Meier survival curves were plotted (Figure 2). All of the low score subgroups showed good prognoses such as EPI^low^ (HR 0.51, 95% CI 0.32–0.81, *p* = 0.0039; Figure 2a), MES^low^ (HR 0.41, 95% CI 0.27–0.62, *p* = 1.4 × 10^−5^; Figure 2b), GLI^low^ (HR 0.27, 95% CI 0.17–0.42, *p* = 1.6 × 10^−9^; Figure 2c), GSC^low^ (HR 0.34, 95% CI 0.21–0.55, *p* = 3.4 × 10^−6^; Figure 2d), MTT^low^ (HR 0.31, 95% CI 0.19–0.51, *p* = 1.5 × 10^−6^; Figure 2e), and PGB^low^ (HR 0.29, 95% CI 0.18–0.48, *p* = 2.7 × 10^−7^; Figure 2f). The CGGA data set also showed similar results, although significances were not detected (Appendix A). These results indicate that the prognosis prediction formulae are useful to evaluate survival in glioblastoma patients. These formulae were associated with strong HRs for the GLI markers (HR 3.70), MTT genes (HR 3.22), and PGBs (HR 3.44), suggesting the success of the formulae. Gene ontology (GO) analysis (Appendix A) indicated the gene sets that were associated with biological processes including proliferation (GO 0008284; *p* = 3.74 × 10^−7^), metabolic process (GO 0051246; *p* = 8.02 × 10^−6^), cell differentiation (GO 0030154; *p* = 4.81 × 10^−5^), migration (GO 0030334; *p* = 7.58 × 10^−5^), and histone H3 deacetylation (GO 0070932; *p* = 1.02 × 10^−4^). These results suggest that the genes used for constructing the prognosis prediction formulae are involved in cancer development and progression.

### 3.4. Genetic Interaction and Network Hubs within the Genes of Interest

We examined the genetic interactions based on the expression intensities to detect the network hubs and central factors in glioblastoma (Figure 3). The rates of composition of interacting to entry genes were relatively high (69.2%–95.8%). The genes constituting the formulae were also included in the genetic network: *CLDN1* in EPI markers (Figure 3a); *CDH11* and *FN1* in MES markers (Figure 3b); *HDAC1/3/7*, *FLT1*, *EGFR*, *PTEN*, *MGMT*, and *PGF* in GLI markers (Figure 3c); *OLIG2*, *STAT3*, *L1CAM*, and *BMI1* in GSC markers (Figure 3d); *HDAC1/3/7*, *SIRT1*, and *EGFR* in MTT genes (Figure 3e); and *GFAP*, *TIMP4*, *IGFBP2*, *MGMT*, and *GADD45A* in PGBs (Figure 3f). Of these, a part of the networks for the biomarkers of MES, GSCs, and PGBs was also replicated in the CGGA data set (Appendix A). These results also suggest that the gene selection in this study was useful for predicting the prognosis of glioblastoma.

### 3.5. Correlation between EMT, GSCs, MTT, and PGBs

Correlations between the six processes were examined using principal component analysis. Each process score was calculated by the sum of the index of the integration of the first principal components and the normalized FPKM values (Appendix C). The correlation between processes was introduced into the scatter plot matrix with correlation coefficient (r) and statistical significances (*p* < 0.001; Figure 4a). Almost all the correlation scores were relatively high (r > 0.61; Figure 4a). The graphical lasso (Figure 4b) detected strong positive correlation between GLI–MTT (r = 0.88) and EPI–MES (r = 0.71), and weak positive correlation between other cellular processes like MES–PGB (r = 0.38), PGB–GSC (r = 0.30), and GSC–GLI (r = 0.25). However, in the CGGA validation data set, these correlation networks were not replicated, such as the negative correlation between EPI and the other processes (Appendix A) and more complex networks drawn in Appendix A. We validated the positive correlation between GLI–MTT (r = 0.87), GSC–PGB (r = 0.43), and MES–PGB (r = 0.43), as shown in Appendix A. These findings suggest that considering MES and GSC as malignant conditions has little meaning in predicting the prognosis of glioblastoma. However, combining GLI with markers of GSCs, MTT genes, and PGBs may enable the development of novel therapeutics and de novo candidate gene markers for predicting the prognosis of glioblastoma.

### 3.6. Assessment using a Combination of EMT and GSCs in the Progression of Glioma

We tested some combinations of EMT and GSC relative to the OS of glioblastoma patients using survival tree analysis (Figure 5). Using the MES and GLI markers, MES^middle^ and MES^low^GLI^high^ showed good prognoses, whereas MES^high^GLI^low^ showed a poor prognosis (Figure 5a), suggesting that the expression of mesenchymal genes could be a good indicator for a prognosis of glioblastoma. MES^middle^ showed a stronger prognosis than MES^high^ (Figure 5b). Using the GLI and GSC markers, GSC^low^ showed a good prognosis, whereas GSC^high^GLI^low^ showed a poor prognosis as compared with GSC^high^GLI^high^ (Figure 5c), suggesting that GSC markers indicate malignant glioma cells. Moreover, PGB^high^ and PGB^middle^ showed poor prognoses as compared with PGB^low^ (Figure 5d,e), thereby validating the candidate markers and formulae used in this study. MES^high^MTT^low^ showed a weaker prognosis than MES^middle^ and MES^high^MTT^high^ (Figure 5f). This suggests that reduced expression of the target genes for MTT leads to poor patient survival. In contrast, the CGGA validation data set returned that EPI^low^ led to a better prognosis than EPI^middle^ and EPI^high^ (Appendix A), suggesting a possibility that the overexpression of EPI genes can distinguish between the initiation of tumorigenesis and normal epithelial differentiation. Important data about GSC markers, MES markers, and PGBs could not be obtained from the CGGA data set (Appendix A). Summarizing these results, the expression of EMT markers including MES, EPI, GSC, and PGBs could indicate a poor prognosis and malignant glioblastoma. Combinations of MES, GLI, and MTT gene markers might be useful for determining the prognosis of glioblastoma.

### 3.7. Multiple Assessments Required for Diagnosis and Predicting Prognosis of Glioblastoma

A combined formula was constituted based on the Cox proportional hazards regression analysis and our previous results (Appendix D). The formula distinguished the Kaplan–Meier survival curves of the subgroups divided by the median scores with a significant HR of 3.08 (95% CI 2.04–4.70, *p* < 0.0001; Figure 6a) and the validation data set with similar results, but were not statistically significant (Figure 6b). To determine the important factors in the prognosis prediction formula based on representative genes like ESI, MES, GLI/GSC markers, and PGBs, we split the formulae used previously into several equations minus various factors (Figure 6c). Interestingly, the formulae without the GSC (ΔGSC) and MES markers (ΔMES) showed the opposite results in the risk ratio compared with the normal combined formula (COMMON) in both the TCGA and CGGA data sets (Figure 6d,e), suggesting that evaluating MES and GSC markers is indispensable for the combined formula. The formulae missing EPI (ΔEPI, ΔEPIΔGSC, and ΔEPIΔPGB) and GLI markers (ΔGLI, ΔGLIΔGSC, and ΔGLIΔPGB) also weakened and/or reversed the risk ratios as compared with COMMON in both data sets (Figure 6d,e). Thus, the appropriate combination of several factors chosen based on the progression of glioma guarantees predicting the prognosis of glioblastoma.

## 4. Discussion

### 4.1. EMT and CSCs in Cancer Development and Progression

Cancer progression involves the induction of genetic stress and EMT, followed by CSC self-renewal, proliferation, and infiltration [38,39,40]. This process is also enhanced by the tumor microenvironment during angiogenesis and evasion of immune response and tumor cell properties, such as proliferation, migration, invasion, and drug resistance [38,40,41]. In glioma, chemo- and radiation-resistant GSCs have properties such as self-renewal and survival that contribute to tumor vasculature by transdifferentiating into endothelial cells upon induction with vascular endothelial growth factors (VEGFs) and hepatoma-derived growth factors (HDGFs) [38]. EMT is associated with infiltration as well as the generation and maintenance of CSCs [39]. EMT also contributes to tumor invasion, heterogeneity, chemoresistance, and robustness in reprogrammed gene expression that enables morphological and functional dedifferentiation into CSCs from differentiated tumor cells [39]. Dysregulation of splicing factors and tumor-specific isoforms frequently occurs in human tumors, indicating the importance of post-transcriptional regulation, including alternative splicing and/or gene fusion [38,42,43]. Thus, it is considered that EMT and stem cell differentiation, including dedifferentiation into CSCs, are closely related to cancer development and progression. Therefore, cellular processes and intrinsic alterations such as alternative splicing would provide the base for developing novel chemical therapeutics and de novo therapies against cancers [27].

### 4.2. Status Assessment and Prognosis Prediction using Multivariable Analyses on Gene Expression Profiling

Increased expression of *PD-L1* correlates with a poor outcome of glioblastoma [44]. Profiling of *PD-L1* expression is useful for determining patient survival time in glioblastoma [19,45]. A lower expression of type 2 T helper (Th2) cell genes compared with Th1 cell genes shows a good prognosis associated with lower expression of *PD-L1* and *PD-1* in glioblastoma, although estimating prognosis based on *PD-L1* and *PD-1* expression is difficult [19]. In the Th2^low^ subgroup of patients with glioblastoma, a reduced expression of solute carrier family 11 (proton-coupled divalent metal ion transporter), member 1 (*SLC11A1*; encoding natural resistance-associated macrophage protein), tumor necrosis factor (TNF) receptor superfamily member 1B (*TNFRSF1B*), and lymphotoxin β receptor (*LTBR*) also show good prognoses [19,46]. TMZ is commonly used for treating glioblastoma and its mechanism of action is well known; however, the side effects associated are largely unknown [47,48]. The TMZ-stimulated glioblastoma cells suppress the expression of pro-inflammatory cytokines in activated periphery blood mononuclear cells. This depends on the enhanced expression of *PD-L1*, but no other immune checkpoint genes, such as *B7-H3*, herpesvirus entry mediator (*HVEM*), and galectin-9 (*LGALS9*), suggesting that TMZ controls the expression of *PD-L1* in glioblastoma cells, resulting in the evasion of the immune system [41]. An increased expression of *B7-H3* and the gene signature comprising *GATA3* and galectin-3 (*LGALS3*) show poor prognoses of glioblastoma [20], suggesting the involvement of the lectin family and oligosaccharide-mediated cell–cell and cell–matrix interactions on the antigen-presenting cell surface in their efficiency to understand glioblastoma progression as well as brain lymphoma [49,50]. Prediction of prognosis based on analyzing cellular processes is applicable to various tumors including other brain tumors, such as primary central nervous system lymphoma (PCNSL). The PD-L1 protein is expressed in tumor microenvironments in 52% of cases, such as immune cells and macrophages. Patients with PCNSL (4.1%) also exhibit a rare and aggressive form of extra-nodal non-Hodgkin’s lymphoma [51,52]. This indicates that PD-L1 expression is detected in the surrounding and tumor cells. Thus, the assessment of cellular processes and prediction of prognosis with statistics from expression data are possible. Differential expression of *PD-1* and *PD-L2* and the Th1/Th2 balance enables the prediction of prognoses of PCNSL [27]. The formula comprising miR-30d, miR-93, and miR-181b is a promising marker for the prognosis of PCNSL [53]. The approaches and methods used in such studies could be applied to various diseases including brain tumors in assessing EMT and CSCs in this study. In addition, the candidates for diagnostic markers may also be detected from comparing with normal brain data.

## 5. Conclusions

In this study, we demonstrated that selective gene marker sets comprising 22 genes (Appendix B), such as *DSG3*, *FN1*, *IGFBP2*, *CLDN1*, *HDAC7*, and *L1CAM* as single gene marker candidates, are useful for predicting the prognosis of glioblastoma, based on the status assessment of EMT and GSCs that can be developed as novel target candidates for therapeutic intervention. The prognosis prediction formulae that included 12 genes (Appendix D), within a mixture of cellular processes, is effective in evaluating the survival of glioblastoma patients, which would help develop novel therapeutics and provide de novo marker candidates to predict the prognosis of glioblastoma.

## Figures and Tables

**Figure 1 cells-08-01312-f001:**
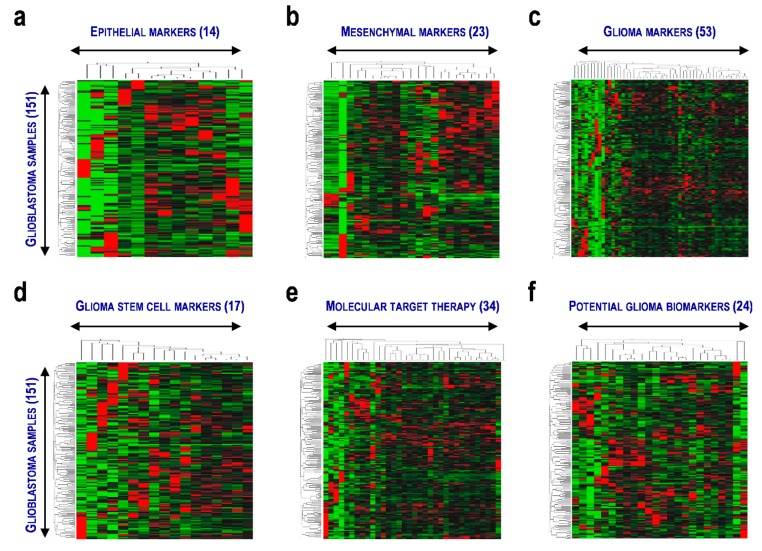
Differential expression of the genes related to epithelial–mesenchymal transition and glioma in 151 patients with glioblastoma. Expression patterns with two-way clustering analysis are presented. (**a**) Epithelial marker gene set. (**b**) Mesenchymal marker gene set. (**c**) Glioma marker gene set. (**d**) Glioma stem cell marker gene set. (**e**) Molecular target therapy gene set. (**f**) Potential glioma marker gene set. Numbers in the parentheses denote the numbers of genes and glioblastoma samples.

**Figure 2 cells-08-01312-f002:**
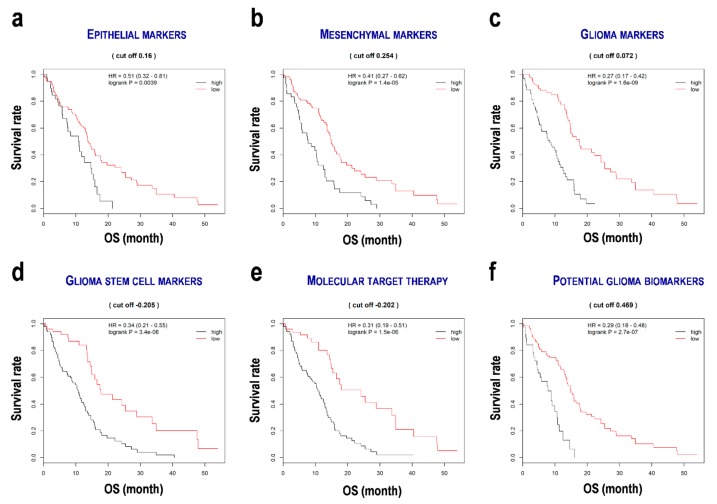
Survival distribution of the subgroups divided by the prognosis prediction formulae in the gene sets in glioblastoma. (**a**) Epithelial marker set. (**b**) Mesenchymal marker gene set. (**c**) Glioma marker gene set. (**d**) Glioma stem cell marker gene set. (**e**) Molecular target therapy gene set. (**f**) Potential glioma marker gene set. OS, overall survival; HR, hazard ratio; cut off score, a median score from a prognosis prediction formula. High and low denote the subgroups of the patients associated with the over and under median scores of the prognosis scores.

**Figure 3 cells-08-01312-f003:**
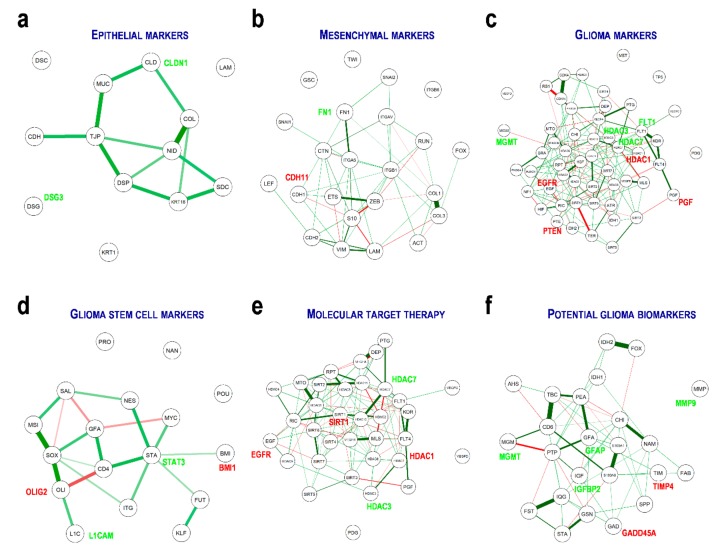
Genetic interaction networks based on the gene expression within the gene sets in glioblastoma. The graphical lasso estimation of the network models is drawn. (**a**) Epithelial marker gene set. (**b**) Mesenchymal marker gene set. (**c**) Glioma marker gene set. (**d**) Glioma stem cell marker gene set. (**e**) Molecular target therapy gene set. (**f**) Potential glioma marker gene set. The genes with significant hazard ratios are highlighted. Red and green represent poor and good prognosis, respectively.

**Figure 4 cells-08-01312-f004:**
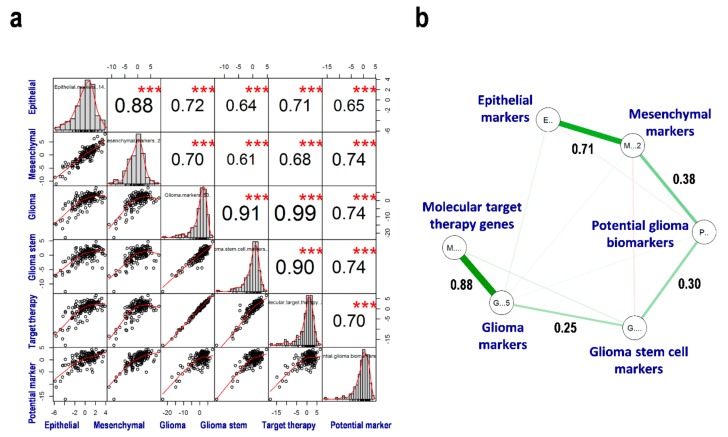
Correlation among the gene sets in glioblastoma. (**a**) Score correlation in glioblastoma. (**b**) The graphical lasso estimation with the network model of the gene sets related to epithelial–mesenchymal transition and glioma in glioblastoma. *** *p* < 0.0001.

**Figure 5 cells-08-01312-f005:**
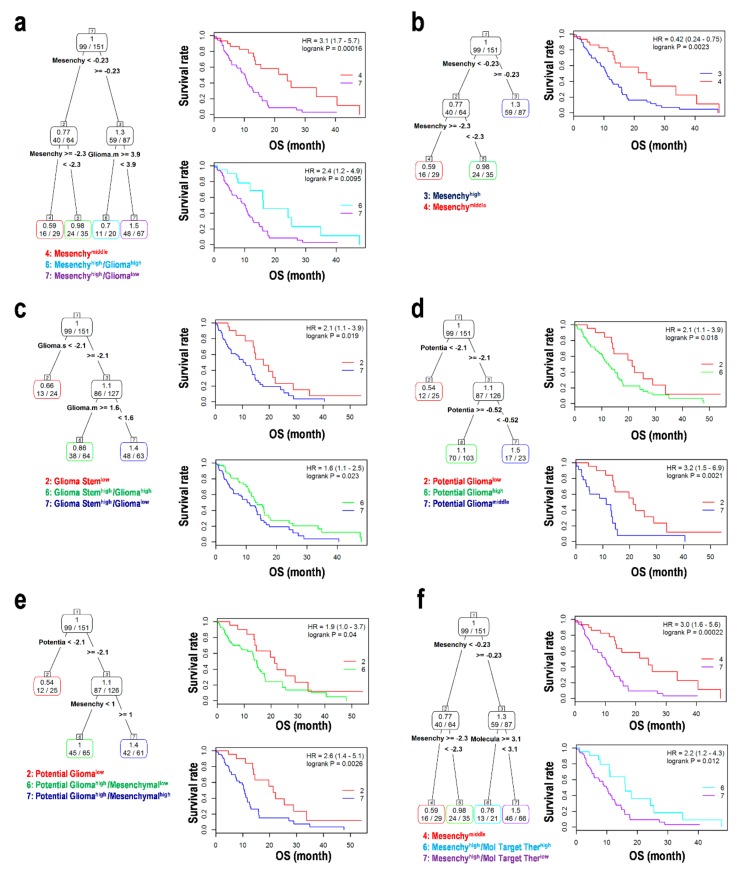
Survival tree analysis for the epithelial–mesenchymal transition (EMT) statuses and glioma markers in glioblastoma. (**a**) Mesenchymal status and glioma markers. (**b**) Mesenchymal status. (**c**) Glioma stem cell markers and glioma markers. (**d**) Potential glioma biomarkers. (**e**) Potential glioma biomarkers and mesenchymal status. (**f**) Mesenchymal status and molecular target therapy genes. HR; hazard ratio, OS; overall survival.

**Figure 6 cells-08-01312-f006:**
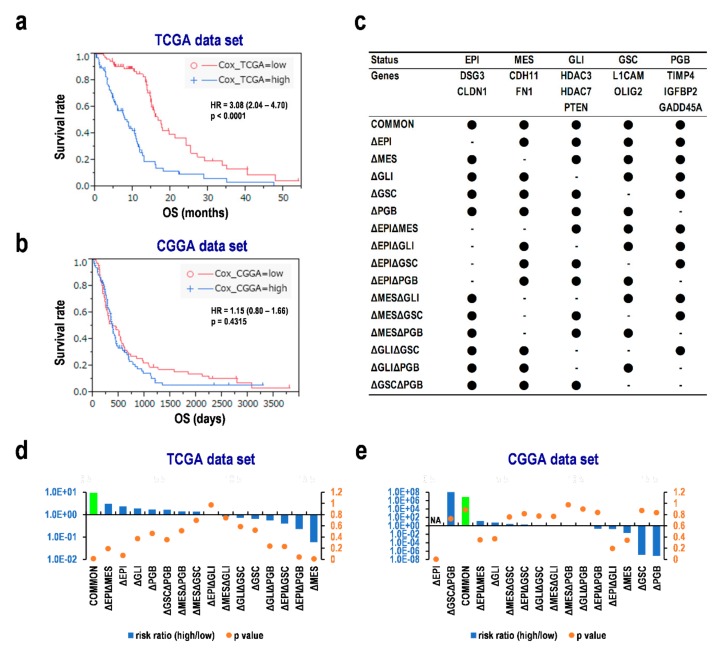
Contributions of EMT status and glioma stem cell markers for prognosis prediction in glioblastoma. (**a**–**b**) Survival distribution estimated with the combinatorial equation for prognosis prediction. (**a**) The Cancer Genome Atlas (TCGA) data set. (**b**) The Chinese Glioma Genome Altas (CGGA) data set. Subgroups were divided by the median score from the formula in the data set. (**c**) The formula composition. (**d**–**e**) The risk ratios and *p*-values in the subgroups with a high score compared with those with a low score. (**d**) The TCGA data set. (**e**) The CGGA data set. EPI, epithelial status; MES, mesenchymal status; GLI, glioma markers; GSC, glioma stem cell markers; PGB, potential glioma biomarkers; NA, not applicable; HR, hazard ratio. Delta (Δ) represents a difference value.

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
