# Peer review of "Promising Prognosis Marker Candidates on the Status of Epithelial–Mesenchymal Transition and Glioma Stem Cells in Glioblastoma"

_cells, 2019, doi:10.3390/cells8111312_

Round 1
Reviewer 1 Report
Since the identification of a gene signature represents an essential aim to improve glioma prognosis and treatment, the manuscript by Yasuo Takashima and collaborators is interesting and current. However some points should be clarified before publication:
-Authors should display a table summarizing the clinical information of patients.
-Since authors evaluated the expression levels of many potential markers by heat maps, they should better explain how they have selected 22 genes to study the overall survival distribution
-Authors declare that the genes used for constructing the prognosis prediction formulae are involved in cancer. Have they an idea about the expression levels of this set of genes in normal brain?
-Authors did not stratify patients in pre-operated and post operated or drug-treated or not treated. Can these parameters influence the prognosis prediction formulae?
Author Response
We are thankful for the reviewers’ constructive comments that helped to considerably improve and clarify the manuscript. We hope that its revised version answers their concerns.
Reviewer 1
Comments and Suggestions for Authors
Since the identification of a gene signature represents an essential aim to improve glioma prognosis and treatment, the manuscript by Yasuo Takashima and collaborators is interesting and current. However some points should be clarified before publication:
-Authors should display a table summarizing the clinical information of patients.
>As suggested, we added Table S1 for clinical information of patients. The related sentences are as follows:
(Page 2, line 79)
The analyses were performed on the TCGA data set (151 samples) and the results were validated by using of the CGGA data set (135 samples) (Table S1).
(Page 11, line 332)
Table S1. Clinical data of patients with glioblastoma examined in the study.
-Since authors evaluated the expression levels of many potential markers by heat maps, they should better explain how they have selected 22 genes to study the overall survival distribution
>Glioma is caused by EMT and maintained by glioma stem cells during tumor progression. Therefore, in this study, we focused on EMT genes and glioma-related genes. The representative genes were selected as previously described. The related sentences are as follows:
(Page 3, line 127)
The aim of this study is to identify promising prognosis marker candidates of glioblastoma. Since glioma is caused by EMT and maintained by glioma stem cells, in this study, we used the expression data and clinical information from 286 glioblastoma patients from TCGA and CGGA (Table S1) and focused on representative genes involved in EMT and glioma (Figure 1, Table S2) [34-37].
-Authors declare that the genes used for constructing the prognosis prediction formulae are involved in cancer. Have they an idea about the expression levels of this set of genes in normal brain?
>The aim of this study is to identify promising prognosis marker candidates of glioblastoma. Therefore, we only used the expression data and clinical information of glioblastoma patients. However, we also consider that the candidates for diagnostic markers may also be found by comparing with normal brain data. The related sentences are as follows:
(Page 3, line 127)
The aim of this study is to identify promising prognosis marker candidates of glioblastoma.
(Page 10, line 315)
In addition, the candidates for diagnostic markers may also be detected from comparing with normal brain data.
-Authors did not stratify patients in pre-operated and post operated or drug-treated or not treated. Can these parameters influence the prognosis prediction formulae?
>Tumors examined were collected from untreated patients during surgery. The related sentences are as follows:
(Page 2, line 81)
Tumor specimens were collected from untreated patients during surgery.
Reviewer 2 Report
In the manuscript several times metastasis is mentioned regarding glioma or glioblastoma. This is a very unusual circumstance with these tumors. I would choose a different word or phrase to discuss their recurrence e.g. infiltration.
Author Response
We are thankful for the reviewers’ constructive comments that helped to considerably improve and clarify the manuscript. We hope that its revised version answers their concerns.
Reviewer 2
Comments and Suggestions for Authors
In the manuscript several times metastasis is mentioned regarding glioma or glioblastoma. This is a very unusual circumstance with these tumors. I would choose a different word or phrase to discuss their recurrence e.g. infiltration.
>As suggested, we replaced “metastasis” to “infiltration”, as page 9, line 271, and page 10, line 277.